# The Landscape of IgA Nephropathy Treatment Strategy: A Pharmacological Overview

Vincenzo Di Leo [1,*,†], Francesca Annese [1,†], Federica Papadia [1], Iris Cara [2], Marica Giliberti [1], Fabio Sallustio [1] and Loreto Gesualdo [1]

1 Nephrology, Dialysis and Transplantation Unit, Department of Emergency and Organ Transplantation, University of Bari Aldo Moro, 70122 Bari, Italy

2 Unit of Obstetrics and Gynaecology, Department of Biomedical Sciences and Human Oncology, University of Bari Aldo Moro, 70122 Bari, Italy

* Correspondence: vincenzo.dileo@policlinico.ba.it; Tel.: +39-080-559-3668

† These authors contributed equally to this work.

**Abstract:** IgA Nephropathy (IgAN) is the most common form of primary glomerulonephritis and is one of the most common causes of end-stage kidney disease (ESKD) worldwide. The immunopathogenic mechanism underlying IgAN is poorly identified. Currently, the mainstay treatment of IgAN is centered on the optimization of blood pressure and a reduction in proteinuria, using an angiotensin-converting enzyme inhibitor (ACEi) and angiotensin receptor blockers (ARBs). According to KDIGO, patients who persistently remain at a high risk of progressive ESKD, despite maximal supportive care, are candidates for glucocorticoid therapy. Recent discoveries regarding the pathogenesis of this disease have led to the testing of new therapeutic drugs targeting, in particular, the excessive mucosal immune reaction and the resulting systemic response as well as the complement activation and the following kidney damage and fibrosis. In this review, we examine the various therapeutic approaches to this intriguing disease.

**Keywords:** IgA nephropathy; new drugs; mucosal hyper-responsiveness; BAFF; APRIL; complement inhibition; kidney fibrosis

## 1. Introduction

In recent years, pharmacological research has produced a multitude of drugs in an attempt to stop the evolution of chronic kidney disease (CKD). In particular, among renal diseases, many advances in the therapeutic approach for IgAN have been made, mainly thanks to the discoveries in understanding the pathogenesis of this rare glomerulonephritis that have been achieved (Figure 1) [1,2].

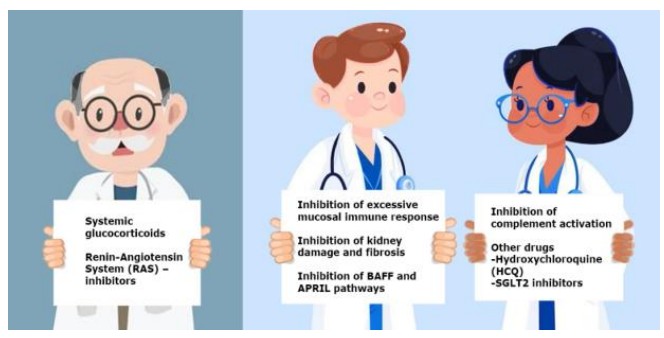

**Figure 1.** Since the initial description of IgAN, significant advances have been made in the nephrological world for the treatment of this rare disease. Here, we report a compilation of selected novel drugs for the treatment of IgAN.

IgAN remains the most common primary glomerulonephritis worldwide [3] and one of the leading causes of ESKD [4]. It is characterized by predominant IgA deposition in the mesangium and is usually accompanied by other immunofluorescence deposits [5].

The glomerular IgA is exclusively of the IgA1 subclass, and it exhibits an incomplete glycosylation (galactose-deficient IgA1; Gd-IgA1) that makes it an autoantigen, inducing the production of autoantibodies [6].

Clinically, the most common manifestation of IgAN is represented by macroscopic hematuria, often concurrent with a mucosal infection of the upper respiratory or gastrointestinal tract. Other patients affected by IgAN presented asymptomatic microscopic hematuria, with or without proteinuria [7].

IgAN is a significant cause of mortality and morbidity in young adults due to the lack of availability of a treatment specific for this particular disease [8]. In fact, until quite recently, the only available therapies for IgAN were inhibitors of the renin–angiotensin system (RAS) and immunosuppressants, especially systemic corticosteroids, although their role is still subject of discussion [9].

By examining the various steps involved in the etiopathogenesis of this disease (Figure 2) [10], this review focuses on the new potential therapeutic weapons against IgAN, while analyzing the ongoing clinical trials.

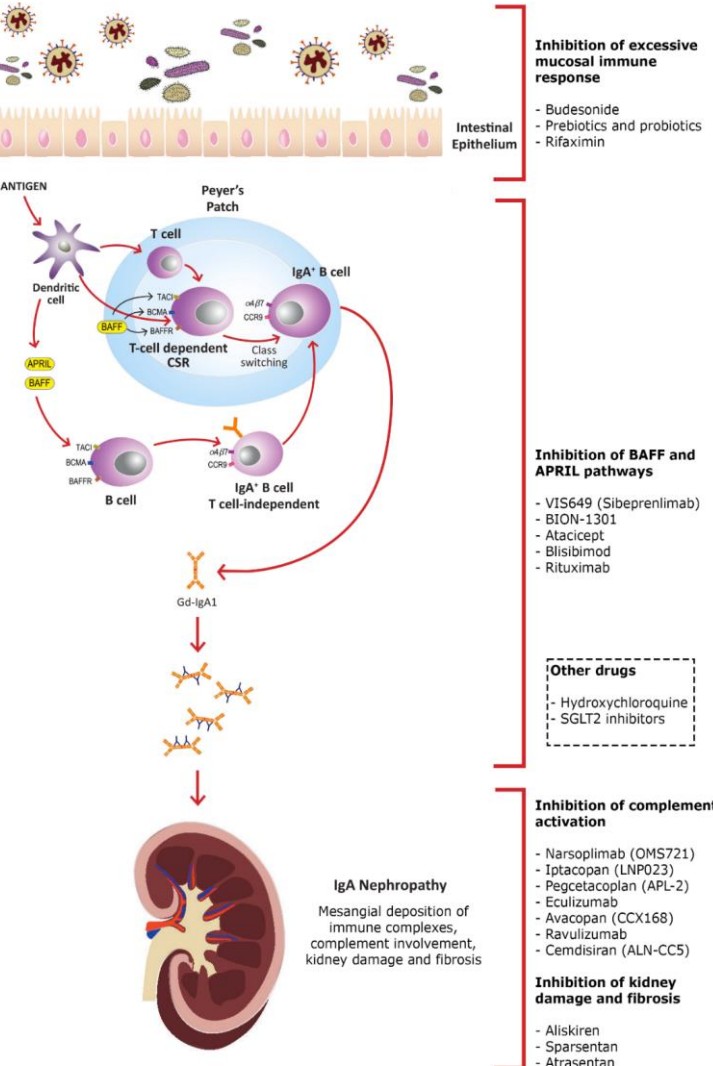

**Figure 2.** Starting from the various steps involved in the pathogenesis of the disease, we report the novel therapies for treatment of primary IgAN.

## 2. Inhibition of Excessive Mucosal Immune Response

Regarding the pathogenesis of IgAN, the "multi-hit" hypothesis seems to be the most widely accepted one. This includes (Hit 1) the production of Gd-IgA1, the formation of IgG or IgA autoantibodies directed against Gd-IgA1 (Hit 2), the formation of immune complexes (Hit 3) and their mesangial deposition (Hit 4), leading to the synthesis of cytokines and chemokines, and the activation of the complement, thus causing kidney damage [6]. Therefore, the process described above suggests that the creation of Gd-IgA1 represents the first step in the development of the disease and that it is actually essential in the pathogenesis of IgAN [11,12].

A central role in the production of Gd-IgA1 is played by the mucosa-associated lymphoid tissue (MALT). The MALT, with the induction of immunotolerance as its main role, comes in close contact with the microbiota and initiates an immune response against environmental microbes [13]. The mucosal immune system appears to be hyperactivated in IgAN patients; indeed, a gut or upper respiratory infection is often accompanied by a new episode of gross hematuria and by an increase in proteinuria [14].

Moreover, modification in the intestinal barrier, with a consequent increased gut permeability [15], is documented in IgAN patients, and it may facilitate an aberrant response to microbial components or alimentary antigens, leading to MALT stimulation and subclinical gut inflammation [16,17].

### 2.1. Budesonide

Due to the central role of the MALT, especially within the Peyer's patches, in the production of Gd-IgA1 molecules, the current treatment of subjects with IgAN is centered around inducing a reduction in the production of these altered immunoglobulins [1]. Indeed, the therapy of IgAN is based on the administration of corticosteroids [18]. Recently, however, some studies have questioned the use of oral cortisone, due to its significant toxic and adverse effects, high risk, and, above all, occurrence of infections [19,20]. In this scenario, the use of the new, oral, targeted-release formulation of glucocorticoid (TRF-budesonide—NEFECON®) has taken hold. This drug is a second-generation synthetic corticosteroid, able to slowly release corticosteroid in the terminal ileum, where there is the maximum accumulation of mucosal Peyer's patches [3]. In the recent NEFIGAN trial and in the subsequent NEFIGARD study, once daily oral administration for 9 months of Nefecon 16 mg showed a statistically significant reduction in proteinuria, a reduced decline in renal function, and dropped serum levels of the Gd-IgA1 and Gd-IgA1-IgG immune complexes [6,21]; moreover, it was also well-tolerated thanks to a specific formulation that allows it to act locally with minimal systemic adsorption [21–23].

### 2.2. Prebiotics and Probiotics

In recent years, the gut microbiota and its metabolites have been shown to have a central role in modulating the immune system's balance [24,25]. The commensal microbes on the gut surface are in contact with the MALT and are implicated in influencing the gut mucosal immune system and gut permeability [26,27]. The close relationship between the microbiota and IgAN has led to a novel therapeutic weapon in the treatment of IgAN through the management of the intestinal microbiota, such as the use of antibiotics, probiotics, and their metabolites or through fecal microbiota transplantation (FMT) [1].

Citing the words of the World Health Organization, probiotics can be described as "live microorganisms which, if administered in adequate amounts, confer a health benefit on the host" [28].

Species from the Lactobacilli and Bifidobacteria genera were revealed to improve humoral immune responses against toxins and antigens. *Bifidobacterium* spp., by its capacity of preventing pathogen invasion, maintaining epithelial and mucosal integrity, and improving host immunity, is well-known to be the most important probiotic in the human body [16]. Moreover, probiotics can produce metabolites by sugar fermentation. These products are

short-chain fatty acids (SCFAs), and the two most well provided in the human colon are sodium acetate (SA) and sodium propionate (SP) [29].

Recently, Qin et al. successfully demonstrated that supplementation with probiotics, including Bifidobacterium, in a mouse model of IgAN might meaningfully relieve intestinal dysbiosis, particularly by increasing the number of helpful bacteria and reducing the richness of potentially pathogenic bacteria. Additionally, both probiotics and their metabolites, SCFAs, might reduce the IgAN symptoms by preventing the activation of the NLRP3/ASC/Caspase 1 signaling pathway [30–32].

### 2.3. Rifaximin

Following the same principle, a new approach to the issue of the abnormally excessive immune response may be represented by the use of antibiotics. In recent research conducted by our team, we demonstrated that treatment with rifaximin, a non-absorbable oral antibiotic, was able to induce a modulation of the intestinal microbiota in a mouse model of IgAN [33,34], promoting the growth of beneficial bacteria. Additionally, binding to the pregnane X receptor promoted the improvement of gut permeability, inhibiting the TLR-4/NF-kB signaling pathway and reducing TNF-α production [35–38]. Moreover, the resulting data showed that the use of rifaximin was associated with decreased proteinuria, a reduction in the serum levels of IgA1-sCD89 and mouse IgG-IgA1 immuno-complexes, and the mesangial deposition of IgA1, thus suggesting a potential place for rifaximin in future clinical trials for this illness [33,37–41].

## 3. Inhibition of B Cell-Activating Factor (BAFF) and a Proliferation-Inducing Ligand (APRIL) Pathways

The pathogenesis of IgAN is significantly influenced by the amounts of Gd-IgA1 and anti-Gd-IgA1 autoantibodies present in the blood. Depleting antibody-producing B cells and reducing pathogenic autoantibodies as a result could be a possible treatment for IgAN.

The TNF family members BAFF and APRIL are inflammatory mediators [42]. They are primarily produced by myeloid cells. However, new research revealed that they are also produced by intestinal epithelial cells, dendritic cells, and local stromal cells and that both hematopoietic and non-hematopoietic cell lineages contain their receptors. Additionally, they are type II transmembrane proteins that are cleaved by proteases and then released as soluble proteins [43,44]. BAFF can be found in both membrane-bound and soluble forms, while APRIL typically only occurs in a soluble form [45]. These two cytokines operate on two receptors: TACI (transmembrane activator and cyclophilin ligand interactor) and BCMA (B cell maturation antigen).

It appears that APRIL binds, particularly firmly and weakly, to both BCMA and TACI, respectively. In contrast, BAFF firmly binds to TACI and to the BAFF-receptor and only weakly to BCMA. Additionally, these three receptors are differently expressed by B cells at various phases of maturation, demonstrating their various functions [46].

In light of the above considerations, targeting the B cell in this particular disease could be a good therapeutic approach because BAFF and APRIL are involved in a variety of immunological diseases, including IgAN, through their role in B cell activation, maturation, and survival [3]. Indeed, when compared to healthy controls, patients with IgAN exhibit higher amounts of APRIL, which appears to be able to stimulate the overproduction of Gd-IgA1 [47].

Recent studies using mouse models treated with anti-APRIL monoclonal antibodies revealed decreased levels of nephritogenic IgA and glomerular IgA deposition. Additionally, sclerosis, mesangial proliferation, and albuminuria were all reduced as a result [48].

Similar to controls, individuals with IgAN were also reported to have significantly higher serum levels of BAFF. The serum levels of BAFF are favorably associated with mesangial hypercellularity and segmental glomerulosclerosis, two indicators of the severity of the histopathological damage [49].

These discoveries offer up new areas of study. In order to induce IgAN remission, researchers are focusing on drugs that can inhibit either BAFF alone or both BAFF and APRIL [12].

### 3.1. VIS649 (Sibeprenlimab)

The fully humanized monoclonal IgG2 antibody VIS649 was the subject of a recent I-phase, randomized, placebo-controlled trial, on which Suzuki et al. recently reported their preliminary findings. The human APRIL cytokine is the focus and antagonist of this molecule's action.

Participants were randomized to VIS649 (sequential i.v. dosing cohorts: 0.5, 2.0, 6.0, and 12.0 mg/kg) or a placebo. According to preliminary findings, there were no serious adverse events (AEs) or AEs leading to study discontinuation, and it was highly tolerated and safe in healthy adults and reversibly reduced serum levels of IgA, Gd-IgA1, IgG, and IgM in a dose-dependent manner. Furthermore, it was demonstrated that following the administration of VIS649, free serum APRIL is reduced to the lower level of assessment [48].

### 3.2. BION-1301

BION-1301 is an additional humanized anti-APRIL polyclonal antibody. BION-1301 was well-tolerated in patients with IgAN, according to interim findings from phase I (NCT03945318) and phase II (NCT04684745) studies reported by Barratt et al. A sustained decrease in blood APRIL levels, as well as decreases in proteinuria and IgA levels, were seen after 12 weeks of treatment with 450 mg every two weeks.

### 3.3. Atacicept

Atacicept and RC-18 are fully humanized fusion proteins that comprise the extracellular region of TACI and inhibit both BAFF and APRIL [3].

Data from a phase II trial comprising patients with more than 1 g/day of IgAN and proteinuria or more than 0.75 mg/mg on a 24 h Urine Protein Creatinine Ratio (UPCR) are available. This was accomplished despite using the maximum tolerated doses of an ACEi or an ARBs [50].

Patients were randomized 1:1:1 to a placebo, atacicept 25 mg, or atacicept 75 mg once per week using subcutaneous injection. A dose-dependent percentage decrease in serum immunoglobulins (IgA, IgG, IgM, and Gd-IgA1) and proteinuria compared to baseline was seen in patients with IgAN after a 24-week treatment period, when atacicept was used instead of a placebo in this study. Additionally, the eGFR levels over time were noted to be stable. This research was prematurely terminated due to slow recruitment, but others on this medication are ongoing.

### 3.4. Blisibimod

A hybrid polypeptide protein made in Escherichia coli, called blisibimod, also known as "A-623," targets both membrane-bound and soluble BAFF [51,52].

The first medication authorized for the management of LES was this monoclonal antibody, a selective BAFF antagonist, and it is now being tested for the treatment of IgAN [53]. According to preliminary findings from a phase II/III study, its subcutaneous administration (induction phase—100 mg three times weekly for 8 weeks, then maintenance phase—200 mg weekly for 16 weeks), when combined with standard therapy, in patients with IgAN showed a substantial decrease in B cell subsets and Ig levels and exhibited pharmacological inhibition of BAFF [54]. Additionally, the blisibimod group, rather than the control, experienced a decrease in proteinuria.

### 3.5. Rituximab

Rituximab is a chimeric murine/human monoclonal anti-CD20 antibody, currently administered as a treatment for some glomerular disorders. However, researchers are currently looking into the proof of its medicinal impact on IgAN [49]. When rituximab binds

to the CD20 antigen that is found on the surface of pre-B and mature B lymphocytes, B cells are lysed [55]. Despite the fact that Lundberg et al. reported several cases of albuminuria reduction and improvement in renal function following rituximab administration, in a randomized, controlled trial based on patients at a high risk of progressive IgAN, Lafayette showed no reduction in albuminuria for 1 year and a positive correlation with several collateral effects per patient [3,56,57].

Rituximab therapy appears to have no effect on serum anti-Gd-IgA1, autoantibody levels, proteinuria, or the rate of decline in renal clearance function in IgAN patients. Last but not least, a controlled, single-blind trial on rituximab is currently underway in China, with preliminary findings anticipated in 2023.

## 4. Inhibition of Complement Activation

The role of the complement system in the pathogenesis of IgAN is not yet well-defined, but it is known to be involved.

In immunofluorescence findings, C3 deposition coexists with IgA in more than 90% of kidney biopsies from patients with IgAN. The histological features of glomerular damage and inflammation, such as mesangial hypercellularity, and the presence of crescents or glomerulosclerosis correlate positively with the abundance of C3 deposits.

The involvement of the alternative and lectin pathway of the complement in IgA pathogenesis is confirmed by the fact that C3, C4, C4d, properdin, mannose-binding lectin (MBL), and the terminal complement complex (C5b-9) are found by immunofluorescence, and the absence of C1q supports the lack of activation of the classical complement pathway. The lectin complement pathway is essential in triggered innate immunity on mucosal surfaces, and IgAN is often characterized by flares during upper airway or gastrointestinal infections. Clinical observations indicate that the activation of the alternative and lectin pathway of the complement might support the pathogenic link between mesangial IgA deposition and kidney inflammation/injury.

As shown by a multivariate analysis of a cohort of 283 patients with IgAN from Spain, there is a relationship between renal C4d accumulation and the progression of ESKD.

This demonstrates that the activation of the lectin pathway influences the severity of IgA-deposited nephropathy early in the development of the disease, when kidney function is still preserved and when there are no severe histological changes in the kidney [58].

Although the serum levels of C3 and C4 are usually in the normal range in patients with IgAN, the suggestion of complement activation is linked with the degree of disease activity and presages a worse kidney outcome.

In patients with IgAN, an elevated IgA:C3 ratio is often found, which is a good indicator to differentiate IgA from other glomerulonephritides, and serum C4 levels are an independent risk factor for disease development. The presence of C5a deposits in the renal sample correlates with proteinuria and the severity of histologic lesions [59,60].

Urinary clearance of complement factors was also recommended as a marker of IgAN activity; indeed, the urinary levels of soluble FactorH and C5b-9 greatly correlate with proteinuria, worse kidney function, interstitial fibrosis, and glomerular sclerosis, while urinary properdin concentrations are only related with proteinuria [3].

Consequently, inhibiting the cascade activity of the complement system could be an important target to improve therapeutic choices.

### 4.1. Narsoplimab (OMS721)

Narsoplimab (OMS721) is a human immunoglobulin G4 (IgG4) monoclonal antibody that binds to and inhibits mannose-binding lectin-associated serine protease (MASP)-2 and blocks the activity of the lectin complement pathway.

Preliminary results of a phase II study (NCT02682407) showed that therapy with OMS721 had no side effects and was associated with a considerable reduction (61.4%) in 24-h albuminuria excretion and a stable eGFR at 31–54 weeks of treatment, in patients with advanced IgAN disease [54].

Currently, in light of these results, a phase III, randomized, double-blind, placebo-controlled trial of narsoplimab in IgAN patients with persistent proteinuria is underway (ARTEMIS-IGAN, ClinicalTrials.gov identifier NCT03608033). The primary objective is to evaluate the effect of OMS721 on proteinuria assessed by 24-h urine protein excretion (UPE) in g/day at 36 weeks from baseline. Enrolled patients must have a diagnosis of biopsy-confirmed IgAN within 8 years of screening, proteinuria >1 g/day, and an eGFR of $\geq$30 mL/min/1.73 m$^2$ calculated according to the CKD-EPI. Throughout the duration of the study, all patients remain in optimized RAS blockade. Patients are randomized 1:1 with OMS721 370 mg or a placebo. The drug is intravenously administered weekly for 12 weeks during initial treatment. Patients are evaluated at T12, and, if UPE > 1 g/day, the patient receives extended treatment (6 weeks).

### 4.2. Iptacopan (LNP023)

Iptacopan (LNP023) is an orally administered targeted factor B inhibitor of the alternative complement pathway. Based on phase II of a clinical trial, patients treated with LNP023 had a 23% decrease in proteinuria compared with the placebo group. In addition, patients in the LNP023 group had a small change in their eGFR, in contrast to an average eGFR reduction of 3.3 mL/min/1.73 m$^2$ in the placebo group [54].

A phase III study (APPLAUSE-IgAN, ClinicalTrials.gov identifier NCT04578834) is now ongoing to explore whether LNP023 might delay IgAN development and recover clinical outcomes. The study is a multicenter, randomized, double-blind, placebo-controlled study to establish the superiority of LNP023 at a dose of 200 mg compared to a placebo on top of maximally tolerated ACEi or ARBs on a decrease in proteinuria and a slowdown in eGFR progression in IgAN patients.

### 4.3. Pegcetacoplan (APL-2)

Pegcetacoplan (APL-2) is a PEGylated, lab-made peptide that inhibits the cleavage and activation of C3, and a phase 2 study is currently underway that is evaluating the security and usefulness of a daily APL-2 subcutaneous infusion administered for 16 weeks with a 6-month safety follow-up, in patients with glomerulopathies such as IgAN, lupus nephritis, primary membranous nephropathy, and C3 glomerulopathy (ClinicalTrials.gov, accessed on 1 March 2023, Identifier NCT03453619).

### 4.4. Eculizumab

Eculizumab is a humanized, recombinant monoclonal antibody that suppresses the release of C5a and the development of the C5b9 membrane attack complex by inhibiting C5 convertase activity. In three case reports, eculizumab was used to treat IgAN. Indeed, two patients with crescentic IgAN were associated with significant renal impairment, despite the use of immunosuppressants treatments, and were treated with eculizumab. This treatment led to clinical improvement with stabilization of the GFR [61,62]. The third case involved the use of eculizumab in a relapse of crescentic IgAN post kidney transplantation with organ failure; in this case, eculizumab was not effective in treating IgAN recurrence after transplantation, probably because therapy was started late, when hemodialysis had already been initiated [58,63].

### 4.5. Avacopan (CCX168)

Avacopan (CCX168) is a drug that specifically inhibits C5aR and reduces the pro-inflammatory and anaphylatoxin effects of C5a, although, unlike eculizumab, it has no effect on C5b9 production. This can protect the innate immune and pathogenic defense functions of the terminal pathway with a lower risk of infection by encapsulated organisms such as those belonging to the Neisserial species. The impact of avacopan on IgAN was studied in a pilot study to assay the safety, tolerability, and effectiveness of CCX168 in reducing proteinuria in IgAN patients with persistent proteinuria despite supportive therapy with maximally tolerated RAS blocker. During the avacopan treatment, six of

the seven patients experienced statistically significant improvement in their UPCR values, three of which had a numerical optimization of ~50% at week 12. At week 24 of therapy, five of the seven patients still maintained this progress in UPCR compared with baseline (ClinicalTrials.gov, accessed on 1 March 2023, Identifier NCT02384317).

*4.6. Ravulizumab*

Ravulizumab, a long-acting humanized monoclonal inhibitor of C5 activation, has similar effects to eculizumab, and it is now being examined in preclinical trials. In addition, a phase II, double-blind, randomized, placebo-controlled study (ClinicalTrials.gov, accessed on 1 March 2023, identifier NCT04564339) is now recruiting participants. This study's goals are to compare the safety and effectiveness of ravulizumab, which is given intravenously (dosages, loading, and maintenance are based on the participant's body weight), compared to a placebo, and establish the efficacy of terminal complement inhibition in subjects with Lupus Nephritis or IgAN.

*4.7. Cemdisiran (ALN-CC5)*

Cemdisiran (ALN-CC5) is a synthetic, small interfering RNA (RNAi) that reduces the hepatic production of C5, and it is being examined in a phase II trial (32-week treatment period, during which patients were dosed with 600 mg of cemdisiran or placebo every 4 weeks) with IgAN patients who are at high risk of progression despite standard of care therapy (ClinicalTrials.gov, accessed on 1 March 2023, identifier NCT03841448).

## 5. Inhibition of Kidney Damage and Fibrosis

According to KDIGO guidelines, the first approach in the management of IgAN patients with proteinuria greater than 1 g/24 h is optimized targeted supportive therapy. This includes lifestyle changes with dietary salt restriction, smoking interruption, body weight control and workouts, and blood pressure control starting by blocking the RAS system with an ACEi or an ARBs (not to be used together), up to the highest tolerated dose.

Angiotensin II (ANG II) directly influences renal hemodynamics by inducing vasoconstriction and causing an increase in the intraglomerular pressure; moreover, it promotes the synthesis of extracellular matrix and cell proliferation, which results in glomerulosclerosis and tubulointerstitial fibrosis. Additionally, ANG II also appears to be implicated in the pathophysiology of podocytes and in the pathogenesis of proteinuria.

Aldosterone also has key roles in the regulation of the sodium/potassium balance and blood pressure, in addition to proscelerotic, fibrogenic, and proteinuric effects.

Endothelin-1 (ET-1) is a vasoactive peptide and, like ANG II, it is implicated in the regulation of renal hemodynamics and has effects on cell growth, extracellular matrix production, and tubular functions.

The two receptors, ETA and ETB, which are located on endothelial cells, mediate the activities of ET-1. ETA and ETB receptors both cause vasoconstriction in vascular smooth muscle cells, but ETB is also expressed on endothelial cells, causing an endothelium-dependent vasodilation.

ET-1 increases the production and activity of proinflammatory and profibrotic signaling molecules and mediators by primarily acting through ETA receptors. The prosclerotic/fibrogenic and inflammatory effects of ET-1 can be a result of ETA receptor activation or as a direct result of ANG II signaling.

Moreover, ET-1 acts as a natriuretic peptide as well, which is different from ANG II or aldosterone, which are both known to have antinatriuretic actions. The collecting duct, which is also the primary location of ET-1's tubular production, is the primary site of ET-1's natriuretic and diuretic activities. One of the aldosterone-sensitive transporters in the collecting duct, the epithelial Na channel, is inhibited by ET-1 through various signaling pathways. Moreover, ET-1 stimulates the Na/H exchanger in the proximal tubule and inhibits the Na-K-Cl cotransporter in the thick ascending arm of Henle's loop, both of

which affect the regulation of the acid–base balance. Furthermore, the peptide decreases water transport that is promoted by arginine vasopressin, improving diuretic effects [64].

### 5.1. Angiotensin-Converting Enzyme Inhibitors and Angiotensin Receptor Antagonists

ACEi and ARBs are the first-line drugs used to treat patients with progressive CKD associated with proteinuria, with the aim of delaying the progression to renal failure. These drugs have a significant nephroprotective effect; in fact, they not only reduce proteinuria but also reduce the rate of glomerular filtration loss. The rationale behind this treatment is that ACEi and ARBs reduce the intraglomerular hypertension that leads to proteinuria through efferent arteriolar vasodilatation of the glomerulus.

Recent studies also showed that ACEi, in addition to improving the pore selectivity of the glomerular basement membrane, can also have positive consequences for mesangial cells by reducing the production of transforming growth factor-$\beta$ (TGF-$\beta$) and by lowering the production of the mesangial matrix, thereby improving the pathological process underlying IgAN.

ARBs was also demonstrated to have an antiproliferative impact on mesangial proliferative glomerulonephritis. All of these discoveries may help to explain why individuals with IgAN who are receiving ACEi/ARBs medication have improved renal function [65].

### 5.2. Aliskiren

It was demonstrated that treatment with aliskiren, the first in a class of medications known as direct renin inhibitors, lowers plasma renin activity, the level of IL-6 or TGF-$\beta$, and the mean UPCR in patients with IgAN.

In a phase 3 study (ClinicalTrials.gov, accessed on 1 March 2023, Identifier: NCT00870493), 22 IgAN patients with persistent proteinuria after maximum therapy were randomized to receive either 300 mg/day of oral aliskiren or a placebo for 16 weeks, and, following a washout period, they switched to the other treatment arm. Following 4 weeks of aliskiren therapy, proteinuria significantly decreased and persisted at a low level for the remainder of the therapy. There were modest but statistically significant decreases in eGFR and diastolic blood pressure following aliskiren therapy [66].

### 5.3. Sparsentan

Sparsentan is a drug that acts as a double blocker on both the angiotensin (AT1) and endothelin (ETA) receptors. The PROTECT study (NCT03762850) is a phase III, randomized, double-blind, active-controlled trial, which aims to determine the long-term nephroprotective potential of sparsentan versus irbesartan in patients with IgAN and is currently ongoing (participants were randomly assigned in a 1:1 ratio to receive sparsentan 400 mg once daily or irbesartan 300 mg once daily).

The interim results show that patients treated with sparsentan have a mean reduction in proteinuria of 49.8% from baseline compared with 15.1% in the irbesartan group [67].

### 5.4. Atrasentan

Atrasentan is a potent antagonist of endothelin receptor A (ETA). Atrasentan was previously demonstrated to benefit subjects with diabetic nephropathy by reducing albuminuria in a study called SONAR, which was designed to evaluate the efficacy and safety of the drug in patients with type 2 diabetes and CKD. Patients were chosen for long-term treatment with this endothelin receptor antagonist based on at least a 30% decrease in UACR and the lack of clinical symptoms of salt retention after short-term low-dose atrasentan therapy. When compared to a placebo, the results demonstrated a substantial decrease in the doubling of serum creatinine or ESKD [68].

The ALIGN study (NCT04573478), a phase 3 clinical trial, is now recruiting patients with biopsy-proven IgAN who have an eGFR of more than 30 mL/min/1.73 m$^2$ and a UPCR of at least 1.0 g/g. All subjects participating in the study will be required to be on a maximally tolerated and stable dose of a RAS inhibitor throughout the study.

Supplementary subjects receiving a stable dose of SGLT2i will be enrolled to the study. The main objective of the ALIGN trial is to assess the impact of 0.75 mg/day of atrasentan on the change in UPCR at week 24. The main objective of the ALIGN trial is to assess the impact of 0.75 mg/day of atrasentan on the change in urine UPCR at week 24 [69].

## 6. Other Drugs

### 6.1. Hydroxychloroquine (HCQ)

In a trial, the effectiveness of HCQ (inhibitor of TLR-9, TLR-7, and TLR-8) was examined in Chinese individuals with IgAN [70]. According to the preliminary findings, taking hydroxychloroquine along with an ACEi or an ARBs for six months reduced proteinuria without affecting eGFR [71]. To validate these findings, additional research in other patient groups is needed.

### 6.2. Sodium-Glucose Cotransporter 2 Inhibitors (SGLT2i)

SGLT2i are a class of medication that acts on proximal renal tubular cells by inhibiting glucose entry through the SGLT2 cotransporter.

They are used for the treatment of diabetes mellitus, but recent studies demonstrated that SGLT2i provides kidney protective effects independently of their antidiabetic's role [72] and that they are a safe addition to the standard medication.

The nephroprotective effects of SGLT2i could be explained by a number of molecular processes such as the control of tubuloglomerular feedback, the transport of sodium to the macula densa, and augmentation of glomerular afferent arteriolar vasoconstriction [72].

In the past few years, three of SGLTi's trials were halted early due to the advantages being so positive. In the CREDENCE study [73], canagliflozin reduced the combined risk of ESKD or death from cardiovascular or kidney-related events by 30%. The drug also significantly slowed the loss of GFR over two and a half years.

In a DAPA-CKD randomized study, 4304 adults affected by CKD were enrolled, and 10 mg/day of dapagliflozin or placebo were administered [68]. Dapagliflozin significantly reduced the onset of ESKD by 36%, the risk of hospitalization for heart failure or death from a cardiovascular event by 29%, eGFR decline, renal impairment, and cardiovascular events (primary combined endpoint). Thus, these beneficial effects were seen both in patients with non-diabetic kidney disease and diabetic kidney disease [3]. Moreover, DAPA-CDK patients randomized to dapagliflozin did not develop diabetic ketoacidosis or major hypoglycemia [74]. It should be emphasized that DAPA-CKD did not include the following varieties of non-diabetic kidney disease: kidney polycystic patients, ANCA-associated vasculitis, lupus nephritis, and patients under immunological therapy for renal disease in the last 6 months. After just two years, the study was stopped due to the obvious advances [75].

A protective effect was shown in trials including individuals with type 2 diabetes, heart failure, and CKD populations, suggesting that SGLT2i may also lower the incidence of adverse events linked to acute kidney injury (AKI) [76].

Furthermore, the EMPA-KIDNEY trial shed more light on the safety of SGLT2i in IgAN patients. A total of 6609 patients were randomly assigned to receive empagliflozin (10 mg once daily) or a matching placebo. During a median of 2.0 years of follow-up, empagliflozin therapy led to a lower risk of the progression of kidney disease or death from cardiovascular causes than a placebo. The rates of serious adverse events were similar in the two groups [77].

The UK Kidney Association (UKKA) guidelines recommend initiating SGLT2i in patients with CKD, heart failure, and type 2 diabetes mellitus (DM) and high atherosclerotic cardiovascular risk. Although the data are not perfectly generalizable due to the selection of participants in each trial, the overall conclusions are clear for people with CKD: the absolute excess risks of amputation and ketoacidosis with SGLT-2 inhibitors are approximately an order of magnitude lower than the absolute benefits of the cardiac and renal outcomes.

There is a particularly low risk of amputation and of ketoacidosis in people without DM, resulting in benefit-to-risk ratios that are particularly favorable in this subgroup [78].

SGLT2i may cause side effects such as genital infections, especially in elderly [79]. Moreover, in the CANAVAS study, is was reported that canagliflozin was associated with an increased incidence of amputations [80]; the EMPA-REG and DECLARE studies, in contrast, did not demonstrate any risk [81]. Fournier's gangrene (necrotizing fasciitis of the perineum) case reports were published, but a causal relationship cannot be established due to the condition's rarity [82].

New potential therapeutic drugs and their respective ongoing clinical trials were shown in Table 1.

**Table 1.** A compilation of new potential therapeutic drugs and their respective ongoing clinical trials for treatment of IgAN.

| Drug | Site of Action | Clinical Research Trial/Study on Mouse Models of IgAN | Possible Outcomes | Possible Adverse Effects (AE) |
|---|---|---|---|---|
| Inhibition of Excessive Mucosal Immune Response | | | | |
| TRF Budesonide | Peyer's patches in the distal ileum | Randomized, double-blind, placebo-controlled study Phase III clinical trial (NefIgArd) NCT03643965 Active, not recruiting | Reduction in UPCR and GFR preservation | Hypertension, peripheral oedema, muscle spasms, and acne [23] |
| Prebiotics and probiotics | Inhibition of NLRP3/ASC/Caspase 1 signaling pathway [32] | 35 patients with IgAN and C57BL/6 mice | Alleviation of gut dysbiosis and attenuation of IgAN clinicopathological manifestations | |
| Rifaximin | Inhibition of microbe-induced immune response and has a direct anti-inflammatory property through binding to the pregnane X receptor (PXR) and modulating gut microbiota [37] | $\alpha 1^{KI}$-$CD89^{Tg}$ mice | Reduction in UPCR, serum levels of hIgA1–sCD89 and mIgG–hIgA1 complexes, hIgA1 glomerular deposition, and CD11b+ cell infiltration | Nausea, stomach pain, dizziness, tiredness, headache, and joint pain |
| Inhibition of BAFF and APRIL Pathways | | | | |
| VIS649 (Sibeprenlimab) | Humanized IgG2 monoclonal antibody that inhibits APRIL | Multicenter, randomized, double-blind, placebo-controlled study Phase III trial (Visionary Study) NCT05248646 [83] Recruiting | Reduction in serum APRIL, IgA, Gd-IgA1, IgG, and IgM | No serious Aes or Aes leading to study discontinuation |

**Table 1.** *Cont.*

| Drug | Site of Action | Clinical Research Trial/Study on Mouse Models of IgAN | Possible Outcomes | Possible Adverse Effects (AE) |
|---|---|---|---|---|
| BION-1301 | Novel humanized blocking antibody targeting APRIL | Part 1: randomized, placebo-controlled single ascending dose design in healthy volunteers; Part 2: randomized, placebo-controlled, multiple ascending dose design in HVs; Part 3 (MD-IgAN): open-label multiple dose design in subjects with IgAN NCT03945318 Active, not recruiting | Reduction in serum levels of APRIL, of immunoglobulins, and in proteinuria | Well-tolerated with no serious adverse events |
| Atacicept | Inhibits BlyS and APRIL [50] | Phase IIb randomized, double-blind, placebo-controlled, dose-ranging study (ORIGIN 3) NCT04716231 Recruiting | Reduction in IgA, IgG, IgM, and Gd-IgA in proteinuria | Aes of special interest included cardiac failure, ischemic heart disease, cardiac arrhythmia, infections, hypersensitivity reactions, and injection-site reactions |
| Blisibimod | Inhibits both soluble and membrane BAFF | Randomized, double-blind, placebo-controlled Phase 2/3 study (BRIGHT-SC) NCT02062684 Completed | Reduction in level of peripheral B cells, immunoglobulins, and UPCR | Upper respiratory tract infection, urinary tract infection, injection site erythema/reaction, and diarrhea |
| Rituximab | Anti-CD20 monoclonal antibody | Multicenter, randomized, prospective, open-label trial NCT00498368 Completed | Changes in proteinuria levels | Fever; cold symptoms, such as runny nose or sore throat; flu symptoms, such as cough, tiredness, and body aches; headache; and cold sores in the mouth or throat |
| Inhibition of Complement Activation | | | | |
| Narsoplimab (OMS721) | Anti-mannan-associated lectin-binding serine protease-2 (MASP-2) | Randomized, double-blind, placebo-controlled Phase 3 study (ARTEMIS-IGAN) NCT03608033 Recruiting | Change from baseline in UPE and proteinuria reduction, rate of change in GFR, and safety and tolerability | Headache, upper respiratory infection, and fatigue |

**Table 1.** *Cont.*

| Drug | Site of Action | Clinical Research Trial/Study on Mouse Models of IgAN | Possible Outcomes | Possible Adverse Effects (AE) |
|---|---|---|---|---|
| Iptacopan (LNP023) | Inhibits factor B inhibitor of the alternative complement pathway | Randomized, double-blind, dose-ranging, parallel-group Phase 3 study (APPLAUSE-IgAN) NCT03373461 Completed | Reduction in UPCR | Headache, abdominal discomfort, blood alkaline phosphatase increase, cough, oropharyngeal pain, pyrexia, and upper respiratory infection |
| Pegcetacoplan (APL-2) | Inhibits C3 protein | Phase 2 study NCT03453619 Active, not recruiting | Proteinuria reduction changes in disease-specific biomarkers (serum C3 levels, AH50 and C3a concentrations, and serum albumin levels) Stabilization or improvement in estimated GFR | No serious or severe AE were reported; stomach pain, vomiting, diarrhea, cold sores, cold symptoms, and tiredness |
| Eculizumab | Inhibits C5 convertase | Case reports [61–63] | Temporary stabilization, but not improvement, of GFR | No AEs were described in the cited case reports |
| Avacopan (CCX168) | Selective C5a receptor inhibitor | Open-label pilot study [84] | Improvement in UPCR | One serious AE of unstable angina, which was deemed to be unrelated to avacopan |
| Ravulizumab | Monoclonal antibody against C5 | Phase 2, double-blind, randomized, placebo-controlled study (SANCTUARY) NCT04564339 Recruiting | Reduction in UPCR and improvement in GFR | Upper respiratory tract infection, diarrhea, nausea, vomiting, headache, high blood pressure, and fever |
| Cemdisiran (ALN-CC5) | Suppresses liver production of C5 protein | Phase 2, randomized, double-blind, placebo-controlled study NCT03841448 Active, not recruiting | Reduction in UPCR | No serious or severe AEs |
| Inhibition of Kidney Damage and Fibrosis | | | | |
| Aliskiren | Direct renin inhibitor | Randomized crossover study NCT00870493 Completed | Anti-proteinuric effect [66] | Hyperkalemia |
| Sparsentan | Selective antagonist of angiotensin II receptor and endothelin A receptor | Randomized, multicenter, double-blind, parallel-group, active-controlled study (PROTECT) NCT03762850 [67] Active, not recruiting | Reduction in proteinuria | Well-tolerated with a clearly defined safety profile |

**Table 1.** *Cont.*

| Drug | Site of Action | Clinical Research Trial/Study on Mouse Models of IgAN | Possible Outcomes | Possible Adverse Effects (AE) |
|---|---|---|---|---|
| Atrasentan | Antagonist of endothelin A receptor | Phase 3, randomized, double-blind, placebo-controlled study (ALIGN) NCT04573478 Active, not recruiting Phase 2, open-label, basket study (AFFINITY) NCT04573920 Recruiting | Effect on proteinuria | Increase in weight and a reduction in hemoglobin [85] |
| Other drugs | | | | |
| Hydroxychloroquine (HCQ) | Inhibits mucosal and intrarenal toll-like receptor signaling | Randomized, double-blind, placebo-controlled study NCT02942381 Completed | Reduction in proteinuria and GFR preservation | HCQ was well-tolerated, and no serious AEs were recorded [71] |
| Sodium-glucose cotransporter 2 inhibitors (SGLT2i) | Modulation of inflammatory and profibrotic mediators and regulation of toxic intracellular compounds (i.e., advanced glycation end products) | International, multicenter, event-driven, randomized, double-blind, parallel group, placebo-controlled study (Dapa-CKD) NCT03036150 Completed Multicenter international randomized parallel-froup double-blind placebo-controlled clinical trial (EMPA-KIDNEY) NCT03594110 Active, not recruiting | Preservation of GFR Reduction in end-stage kidney disease, in death from renal causes, in death from cardiovascular causes, or inhospitalization for heart failure | Genital infections Changes in urination, including urgent need to urinate more often, in larger amounts, or at night [79] |

## 7. Conclusions

Since the initial discovery of IgAN, important progress has been made in understanding the development of the disease. The most accredited hypothesis about the pathogenesis of IgAN is the "four-hit" theory, which underlines the centrality of immunological factors in all aspects of IgAN development. These advances have led to the development of new and promising therapeutic strategies. In fact, there are several encouraging novel medications for IgAN in their earlier stages of clinical development that are driving the treatment of this disease in the direction of personalized medicine.

Starting from the various steps involved in the pathogenesis of the disease, this review aims to explore innovative therapeutic strategies in order to improve the outcome for IgAN patients.

**Funding:** This research received no external funding.

**Institutional Review Board Statement:** Not applicable.

**Informed Consent Statement:** Not applicable.

**Conflicts of Interest:** The authors declare no conflict of interest.

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
