# Peer review of "The Landscape of IgA Nephropathy Treatment Strategy: A Pharmacological Overview"

_futurepharmacol, doi:10.3390/futurepharmacol3020033_

Round 1

Reviewer 1 Report

Comments and Suggestions for Authors

The proposed review on the current advances in the IgA nephropathy options of treatment is well documented and clearly structured. Therefore, I have only minor comments:

1. A reference to Figure 2 (which is very informative) could be made in the body text earlier. I suggest to insert it from the Introduction (on page 2, either paragraph 2 or 3).

2. The review lists almost a large number of potential medications tested for IgA nephropathy. However, for the practical point of view it would be helpful to highlight the research stage for every drug and the perspective of its future utility. For this aim, I suggest to add a summarizing table including a column with clinical applicability of the tested drugs.

3. In line with the previous comment, the authors should emphasize the central role of SGLT2 inhibitors in the light of the latest published clinical trials. As a consequence of these, there are already guidelines which suggest to introduce this class of drugs in the routine management of patients with non-diabetic CKD (UKKA guidelines). The authors could also include the criteria for prescribing SGLT2i in this category of patients, as well as some considerations about their adverse effects.

4. Short comments on adverse events should be added for the other therapeutic classes as well.

5. The paragraphs related to sparsentan (page 9) lacks a reference. Please verify and add the corresponding one.

Author Response

Dear reviewer, thank you for your comments.

  • We inserted the reference of figure 2 in the “Introduction” (paragraph 1).
  • We created a table in witch we summarized the tested drugs, in particular we specified for each drug the site of action, the kind of clinical research trial or if the drug is used only on a mouse model, the possible outcome and the possible adverse events.
  • We changed the paragraph in which we described the SGLT2 inhibitors, including the UKKA guidelines, the criteria for the prescription and also their adverse effects.
  • Please, find it in the table.
  • We added a specific reference.

Reviewer 2 Report

Comments and Suggestions for Authors

In introduction, Add more literature about nephropathy and then about immunogenic nephropathy and reason for this type of nephropathy.

Author need to start from nephropathy, reasons of nephropathy and then types of nephropathy. After this explain about immunogenic nephropathy. literature about pathogenesis of immunogenic nephropathy need to be added

It will be good to draw a pictorial diagram to show pathogenic factors of immunogenic nephropathy

write full form of BAFF and APRIL

Fig. 2 should be cited in discussion, not in conclusion

Author Response

Dear reviewer, thanks you for your kindly reply.

  • Following you suggestion, we added more literature about the IgA nephropathy. Moreover we reported the single steps involved in the pathogenesis of the disease in each corresponding paragraph.
  • We changed the title of the paragraph 3: Inhibition of B cell-activating factor (BAFF) and a proliferation-inducing ligand (APRIL) pathways
  • We cited “Fig. 2” in the “Introduction”

Reviewer 3 Report

Comments and Suggestions for Authors

The authors gave an overview of past, present and future therapeutic strategies in the treatment of IgA nephropathy. The text is well structured however I would recommend the following changes: 1.many parts of the work (especially in the paragraph on angiotensin, aldosterone and endothelin) have a structure and phrasing too similar to the cited articles: I would ask the authors to make everything more personalized. 2.I would ask for the paragraphs relating to the drugs to be standardized: for some there is an extensive description with timing and dosage, for others only hints. 3. I would create a separate short paragraph on pathogenesis right after the introduction, quoting figure 2 (nicely done) first. 4. Figure 1, nice and schematic, I would see more as a visual abstract rather than part of the article. 5. The paragraph on Eculizumab mentions the prognosis but only on the third patient: how was the prognosis of the other two?

6.The sentence on vasoconstriction and vasodilatation of ETB is not clear: in fact, according to the reference, ETA and ETB both mediate vasoconstriction when expressed in vascular smooth cells, but ETB is also expressed on endothelial cells, resulting in a vasodilatory effect.

Author Response

Dear reviewer, thanks you for your kindly comments.

  • In order to avoid similarities with other article, we used “Plagiarism Checker”
  • We added the timing and dosage for all of the drugs
  • We added other information about the pathogenesis in the introduction. Moreover we explored the pathogenesis of the disease in each corresponding paragraph.
  • I have to ask to the Editor if the visual abstract could be contemplated.
  • We changed the paragraph.
  • We changed the sentence.

Reviewer 4 Report

Comments and Suggestions for Authors

This manuscript is a narrative review of current and potential novel treatment strategies for IgA nephropathy. The manuscript is well-structured and comprehensively described.

Minor comments

I suggest changing part of the title of the manuscript. Instead of a pharmacology overview, I suggest a pharmacological overview or a pharmacologic overview.

On page 6, part 4. Inhibition of complement activation, I suggest normal range instead of standard range. This term is usually used in clinical practice.

On same page for the sentence "The presence of C5a deposits in the renal sample correlates with proteinuria and the severity of histologic lesions." a reference lacks to confirm that statement.

On page 8, grams are expressed as gr instead of g. This should be corrected throughout the manuscript.

On page 8, the authors use an incorrect name/term for the renal collecting duct. Instead of a collector duct, it is correct term a collecting duct.

On the page 9, in the text "In a trial, the effectiveness of HCQ ((inhibitor of TLR-9..." ( is redundant.

The reference for the EMPA-KIDNEY study is missing.

Author Response

Dear reviewer, thanks you for your kindly comments.

  • We changed the title
  • We changed normal range instead of standard range
  • We inserted references about "The presence of C5a deposits in the renal sample correlates with proteinuria and the severity of histologic lesions."
  • We changed g instead of gr
  • We changed collecting duct instead of collector duct
  • We erased “(“
  • We cited and reported EMPA-KIDNEY study

Round 2

Reviewer 3 Report

Comments and Suggestions for Authors

Well done!

I have no further suggestions to ask.